# Determinants of Healthy and Active Ageing in Korea

**DOI:** 10.3390/ijerph192416802

**Published:** 2022-12-14

**Authors:** Alexandre Repkine, Hyun-Chool Lee

**Affiliations:** 1Department of Economics, Konkuk University, Seoul 05029, Republic of Korea; 2Department of Political Science, Konkuk University, Seoul 05029, Republic of Korea

**Keywords:** healthy and active ageing, senior citizens, senior employment

## Abstract

Based on a framework developed by the World Health Organization, we construct an individual-level percentage measure of healthy and active ageing employing the results of a unique survey of ten thousand elderly Korean respondents conducted in 2020 and relate its values to the senior respondent’s physical, lifestyle, and socio-economic characteristics. We find that the median value of our healthy and active ageing index is approximately 40%, suggesting significant room for improvement. An important role in interpreting our empirical results is played by the apparent role of Korea’s senior employment as a means of “making ends meet” rather than a way of improving the quality of one’s ageing, suggesting an important direction for government policy development. Our results underscore the importance of promoting higher-quality employment opportunities for senior citizens as opposed to creating these opportunities per se. This appears especially important given the fast pace of Korea’s ageing.

## 1. Introduction

The population in South Korea (henceforth Korea) is ageing rapidly, necessitating the development of the associated measures of the quality of ageing and the determinants thereof. Indeed, Korea officially became an ageing society in 2000 with the share of the older population, i.e., senior citizens aged 65 and above, reaching 7% and the total fertility rate (TFR) falling below 1.3 by 2002, well below the TFR level of 2.1 needed for reproduction, see [1,2]. It took Korea only 17 years to become an aged society with over 14% of her population older than 65 in 2017 as opposed to Japan where the same transition took 24 years, see [3]. Meanwhile, as evidenced in [4], Korea’s TFR fell to 0.81 in 2021, the world’s lowest level. According to [5], in the year 2020 Korea officially registered a first-ever decrease in its population. By 2025 Korea will likely become a super-aged society with the share of older population exceeding 20%, as projected by KOSTAT in [6]. As of 2021, four regions in Korea have already reached the super-aged status, i.e., the provinces of North and South Jeolla, North Gyeongsang, and Gangwon, see [5].

Several factors may account for the rapid ageing of the Korean population. Rising apartment prices and increased costs of children’s private education often deter young Koreans from marrying at a younger age. Increased coverage of Korea’s older population by medical insurance and improved general living standards resulted in a significant increase in the older generation’s life expectancy, as documented in [2]. Thus, in 2019 life expectancy at the age of 65 stood at 21.3 years, which is 1.4 years higher than the OECD average, as reported by [6].

Korea’s ageing population brings forth a host of well-known problems, such as, e.g., a shrinking labor force or a higher dependency ratio, see [2]. The latter is measured as a number of senior adults per one working individual and is projected to exceed 70% by the year of 2049, see [7]. Reduced tax revenues due to the shrinking labor force will not only result in lower spending on welfare programs for the elderly but also in the pension fund deficits, dealing another blow to the well-being of the older generation, as argued in [2]. The Korean Development Institute (see [8]) predicted that the growth rate of the Korean economy would gradually slow down after the 2020s due to demographic changes, such as population decline and rapid aging, so the economic growth rate would drop to 0.5% in 2050.

Korea’s population ageing necessitates the government’s policy response, such as, e.g., developing fiscal policies aimed at establishing healthcare models that are specially designed for older individuals or policies promoting productivity improvements that are necessary to make up for the rapidly declining labor force, see [9]. In addition, the creation of opportunities for older citizens’ productive contribution to society is necessary, such as, e.g., vocational training and lifelong education programs. Immigration-related policies could play a positive role similar to the examples of the Turkish immigration to Germany or the Poles to Belgium in the middle of the twentieth century, as argued in [10].

No matter what the Korean government’s policy responses might be, they will be more effective in case the ageing population is healthy and active as recognized by the European Commission (EC) in [11] in its attempts to develop a multi-dimensional concept of active ageing, which is also referred to as healthy ageing, with an important contribution made by the World Health Organization (WHO), see [12]. The indices of healthy and active ageing developed by EC and WHO were initially computed for the European Union economies, but the methodology was later extended to other countries as well due to the importance of measuring the quality of ageing, as the governments need to design policies dealing with the ageing-related problems outlined above, see, e.g., [13] for an application to the case of Cameroon.

In this study, we follow the recommendations in [14] in order to compute an individual-level index of healthy and active ageing (henceforth HAA index) on the basis of a rich and unique National Survey of Older Koreans covering more than ten thousand individuals, conducted in 2020 and summarized in [15].

To our knowledge, a comprehensive index of healthy and active ageing developed in accordance with the WHO framework has not been computed in Korea so far on the basis of a country-wide national survey sample. One purpose of this study is thus to provide policy makers with a quantitative idea of the structure of ageing in Korean society that can also be analyzed from an international perspective. Second, we conduct a rigorous quantitative analysis of the physical, lifestyle, and quantitative characteristics of senior individuals that affect the quality of their ageing. We hope the results of this study are valuable not only in terms of the academic contribution but also in terms of their potential applicability to policy making.

We find that there exists significant room for improvement in the quality of ageing in Korea as measured by the HAA index. Our analysis of the HAA index determinants suggests that at least for now the government policy should focus more on the determinants related to the quality of employment of senior citizens in Korea.

This study is organized as follows. In Section 2 we review the literature related to the measurement of the quality of ageing, focusing on the efforts by the European Commission and the World Health Organization, and identify the likely determinants of healthy and active ageing. We start Section 3 by describing the design of the National Survey of Older Koreans used as a data source for this study. We proceed by detailing the computation of the individual-level HAA index and conclude Section 3 by describing the logit and probit approach to analyzing the association of the HAA index values and its determinants. The empirical results are presented in Section 4 with Section 5 discussing them. Section 6 is the conclusion.

## 2. Literature Review

### 2.1. Indices of Healthy and Active Ageing

While there is no doubt that, given the rapidly ageing world population, the governments should devise policies promoting the older generation’s ability to age successfully, there has been a considerable disagreement by the scholars on how to define the concept of successful ageing, see, e.g., discussions in [16,17]. As noted by [18], even the name of the concept has not been quite established as a cursory examination of the related literature would suggest. Thus, apart from successful ageing, various other terms were used such as, e.g., active, productive, or positive ageing. In their seminal work [19] conceptualize the idea of healthy ageing. the European Commission in [11] developed and operationalized a multi-dimensional concept of active ageing. In [13] the authors use the term “healthy and active ageing”.

The earlier approaches to defining and measuring quality or successful ageing were biased towards health or physical characteristics of older individuals. Thus, [19] models successful ageing as opposed to the normal ageing based on the four domains, namely, (1) the absence of seven major diseases and their associated risk factors, (2) absence of physical disabilities, (3) high physical function, and (4) high cognitive function and engagement in life. A similar four-dimensional approach is pursued in [20,21]. There increasingly emerges an understanding of the need to abandon a uni-dimensional (e.g., physical characteristics-based) approach in favor of a more inclusive, multi-dimensional conceptualization, see a general discussion in [22] and a discourse on, e.g., Korean older voters’ political participation in [23].

An important contribution was made in this regard within the framework on active ageing presented by the World Health Organization (WHO) in 2002 in that it explicitly recognized the important role played by older individuals’ participation in the society and their economic and physical security. Thus, the definition in [12] says: “*Active ageing is the process of optimizing opportunities for health, participation and security in order to enhance quality of life as people age*”. This definition was later modified to “*an ongoing process of developing and maintaining the functional ability that enables well-being in old age*”, see [24]. In other words, there occurred a shift from an understanding of successful or quality ageing that is based on physical characteristics towards an evaluation based on the functional outcomes that related to the extent of older individuals’ participation in the society, economic security, and capacity for increasing the quality of their ageing.

As a result of the efforts to operationalize the idea of a multi-dimensional concept of an index capturing various aspects of successful or quality ageing, the European Commission in [14] developed a set of guidelines on the computation of what they termed an active ageing index, or AAI. Figure 1.1 in [14] suggests measuring active ageing by using twenty indicators belonging to four broadly defined domains, namely, (1) Employment, (2) Participation in Society, (3) Independent, Healthy and Secure Living, and (4) Capacity and Enabling Environment for Healthy Ageing. An important distinction of the European Commission’s approach to measuring the quality of ageing from the previous attempts is to focus on the outcomes rather than the inputs, or processes. To cite [11], “*the index shows the actual activity of the current generation of older people through which they contribute to the economy and society*”.

As a result of this outcome-based approach, the AAI does not include information on the older individuals’ chronic diseases as these are processes or inputs associated with ageing as opposed to the outcome thereof. Indeed, while diseases and illnesses such as, e.g., arthritis or high blood pressure are definitely altering the way one is ageing, they do not necessarily lead to the deterioration of the ageing outcomes. Thus, a person diagnosed with high blood pressure may still maintain good physical health by combining a proper diet with physical exercise, as mentioned in [11]. At the same time, the outcome-based approach creates certain problems as the distinction between inputs and outcomes is sometimes not obvious. For example, physical exercise is an indicator within the “Independent, Healthy and Secure Living” domain, but it is also an input to one’s ability to age successfully, actively, and healthily. In this study we follow the classification developed by the European Commission in computing a version of the AAI adapted for computation at an individual level.

Several studies attempt to evaluate the quality of ageing in Korea by adapting the AAI concept to the Korean cultural and economic realities. Thus, [25] develop a version of the AAI for Korea and compare its value with that of China and the average for the European Union. An index of healthy ageing for Korea based on three domains each comprised of eleven indicators was developed by [26]. Similar to this study, the author uses the results of a comprehensive survey in order to evaluate his measure of healthy ageing. Differently from the AAI developed by the European Commission, however, his index is calculated at an individual, rather than aggregate, level. A multidimensional approach was pursued in [27] to relate various measures of life satisfaction among Korea’s older population to a number of socio-economic and physical characteristics, although the authors do not construct any single value characterizing the extent of life satisfaction among the older Koreans. Four nested definitions of healthy ageing based on the model developed in [19] were used in [28] in order to analyze the association between healthy ageing and mortality risks of older Koreans.

In this study we develop an individual-level healthy and active ageing index based on the four-dimensional approach by the European Commission in [14] and relate it to a set of physical, lifestyle, and socio-economic determinants. In line with the general division made by the EU framework we relate healthy ageing outcomes to a set of determinants that can be viewed as inputs to, or processes associated with, the process of ageing, rather than its outcomes.

### 2.2. Determinants of Healthy and Active Ageing

One comprehensive review of the literature on the association between various measures of healthy ageing and a set of determinants is [22]. Income, wealth, education and employment status are rather expectedly found to be positively associated with the likelihood of healthy ageing. Similarly, [13] consider the effects of education and income on several dimensions of healthy ageing with a surprising finding of no statistically significant association between education and healthy ageing. The problem with this group of studies is, however, that these determinants can as well be considered to be outcomes, rather than the determinants, of healthy ageing. Thus, education is a component of the “Capacity and Environment” domain of the active ageing index developed in [14], income is a factor contributing to the domain “Independent and Healthy Living”, and employment status is a defining characteristics of the domain “Employment” in the EU framework. 

It appears to be more reasonable to look at those physical, lifestyle and socio-economic characteristics of older individuals that are not considered to be components of healthy and active ageing at the same time. A lifestyle index comprising the extent of physical activity, consumption of vegetables and fruits, regular consumption of meals and adequate consumption of liquids is found to be positively related to healthy ageing in [29]. Except for the physical exercise that is a component of “Independent and Healthy Living” domain in the AAI definition of EU and WHO, all components of this lifestyle index make sense as determinants of healthy and active ageing. A comprehensive review of the determinants of healthy ageing is identified in the existing literature by [30] that includes physical activity, diet, self-awareness, life-long learning, faith, social support, financial security, community engagement, and independence. Of these, diet and faith are the only two determinants of healthy ageing that do not enter the set of the AAI indicators defined in [14].

In this study, we make a careful distinction between inputs and outcomes of the ageing process so as not to include the components of the healthy and active ageing index we describe in Section 3.4 in the list of its determinants. For example, while we take education to be a determinant of the domain “Employment” of our healthy and active ageing index, it does not enter the list of the determinants for the comprehensive index that captures all of the four domains. For the same reason an individual’s income is excluded from the list of determinants of the domain “Independent and Healthy Living”. In this study we consider the following groups of determinants affecting the value of healthy and active ageing: physical characteristics such as, e.g., weight or gender, lifestyle variables such as smoking or eating enough vegetables, and socio-economic characteristics such as the married status, receiving a public pension, or housing type. As mentioned above, we make sure none of these determinants are double accounted for as components of the outcome variables in our analysis. The complete list of variables used in the statistical analysis is presented in Section 4.3.

## 3. Materials and Methods

### 3.1. National Survey of Older Koreans and Its Design

Our analysis is based on the results of a 2020 National Survey of Older Koreans [15] covering 10,097 Korean respondents aged 65 and above commissioned by the Korean Institute for Health and Social Affairs. While traditionally the age of 60 used to be considered the start of old age in Korea, more recently the age of 65 became widely recognized as an old age threshold used for the design of the pension benefits schemes, the system of public transportation discounts and the like. The survey employed in this study is the fifth in a series of surveys that are conducted once every three years starting in 2008.

The questions of the survey were designed by 28 experts in several fields. The survey’s questions were scrutinized by the members of the Institutional Review Board that forms part of the Korean Institute for Health and Social, receiving an official approval. The survey was conducted by means of the Tablet-PC Assisted Personalized Interviews. 

Since February 2013, for the purpose of protecting research subjects in accordance with the ‘Bioethics and Safety Act’, the establishment of an Institutional Review Board (IRB), an autonomous review body within research institutions, has been mandated in order to review the scientific and ethical validity of research protocols. Accordingly, the 2020 Elderly Survey applied for IRB for research plans and other research-related documents to the Bioethics Committee established within this institution, and after review, IRB approval (Korea Institute for Health and Social Affairs Bioethics Committee (IRB) Review Result Notification No. 2020-36) was received.

### 3.2. Survey Sample Design and Sample Characteristics

The stratification of the sample was based on the Korean Population and Housing Census conducted by the Korean National Statistical Office that details population numbers in 17 Korean city and provincial regions, namely, seven metropolitan areas such as Seoul or Busan, nine provincial areas, and a special region of the Jeju island. The survey sample was designed in order to reflect the shares of older population residing in these 17 areas, and the shares of the older population residing in the apartment buildings within these areas.

For the sample distribution by dong/eup/myeon in each city/province, the proportional distribution method based on the number of elderly people aged 65 or older in the population and a housing census was applied. The results of sampling distribution by dong/eup/myeon in each city/province are shown in Table 1 below:

### 3.3. Missing Observations

Due to the fact that some responses were missing the sample on the basis of which we are conducting our statistical analysis has 9306 respondents compared to the original one containing 10,097 individuals. The value of the $\chi^{2}$ statistics for the Little’s test of the data missing completely at random (MCAR, see [31]) is estimated at the level of 1309, which implies we cannot reject the hypothesis of the missing data being completely independent of the observed data points. However, since the number of the individuals with missing responses on one or more variables is less than 8% of the original sample, the results of a statistical analysis based on a reduced sample obtained by listwise deletion (i.e., ignoring those respondents for whom one or more responses are missing) is unlikely to be strongly biased, see [32]. In addition, the sample size of 9306 is still larger than the size of 8943 recommended by the simple random sampling approach and calculated according to (1), and is large enough to allow for the derivation of meaningful statistical conclusions.

### 3.4. The Computation of Individual Index of Healthy and Active Ageing

In this study we analyze the determinants of healthy and active ageing at an individual level on the basis of an index of healthy and active ageing (henceforth HAAI) that we compute employing the WHO policy framework for the evaluation of active ageing at an aggregate level. The original active ageing index developed within the WHO framework, see, e.g., [12], consists of twenty-two indicators belonging to four domains, namely, Employment, Participation in Society, Independent and Healthy Living, and Enabling Environment for Healthy Aging. All of these indicators represent groups of populations. For example, indicator 1.2 in the Employment Domain is the employment rate for the group of individual aged between 60 and 64. Indicator 3.4 in the independent and healthy living domain is the relative median income in the group of individuals older than 55.

Due to the fact that we are attempting to measure the HAAI at an individual, rather than a group, level, we produce two necessary adjustments to the WHO methodology. First, we replace group indicators with their individual analogues. Thus, the median income indicator becomes an older respondent’s income relative to the nation’s median. In case of the Employment domain that consists of the employment rates of four different age groups, we represent it with one binary score reflecting the individual’s employment status. In addition, we choose to refer to our index as “healthy and active ageing index” both in order to distinguish it from the aggregate index of active ageing developed by the WHO and in order to emphasize the fact that the quality of ageing is depending on a combination of both physical health and active participation in the society, as stressed by [19].

In addition, we produce an important adjustment regarding the weights pertaining to both the four domains and the individual indicators within these domains. The WHO framework assigns explicit and implicit weights to these domains and indicators. Implicit weights are the ones assigned to domains and their constituent indicators by a group of experts in the related fields, see [11]. Explicit weights are computed as adjustments to the implicit weights in order to compensate for the differences in the magnitudes of the indicators’ values. This adjustment, however, is not necessary in case all indicators are standardized to vary within the [0, 1] interval, which we do in our study. As a result, we are using the implicit weights recommended by experts and reported in [11] in constructing the HAAI. In case of the third and fourth domains where we exclude some of original indicators (see Section 3.4.3 and Section 3.4.4 below), we adjust the remaining weights upward to make sure they sum up to unity, while keeping their relative magnitudes intact.

Table 2 below represents the composition of the individual healthy and active ageing index that we are employing in this study.

We follow the WHO recommendation to evaluate the quality of ageing based on the four domains each consisting of a group of indicators, see Table 2, with the total number of indicators being 17. As the second row of Table 2 suggests, the first three domains, i.e., Employment, Social Participation, and Independent and Healthy Living, are capturing the actual experience of ageing, while the fourth domain, the Capacity and Environment for Healthy and Active Ageing, is reflecting those factors that are contributing to the realization of the first three domains.

#### 3.4.1. Employment Domain

As discussed above, we represent this domain with a single binary indicator that assumes the value of one in case of the 37.5% of the individuals who reported working for income for at least one hour during the week when the survey was taken, in line with the recommendations in [11]. The remaining 62.5% receive the score of zero. The employment domain enters the HAAI index with the weight of 28%.

#### 3.4.2. Social Participation Domain

In the Social Participation domain we keep all of the four original indicators as these can be easily adapted to the individual-level computations. For instance, indicator 2.1 (Voluntary Activities) is originally the share of respondents engaged in the voluntary or charity work. We use a binary variable that is equal to one, if a respondent reports being involved with the voluntary activities. The same goes for the indicators capturing the extent of help to children, grandchildren, and older parents. We extend this domain to accommodate questions related to helping an older respondent’s spouse.

In our survey there are a series of questions related to the help extended by older individuals to the members of their families within the course of the last year. The answers are Likert-type on a scale from 0 to 4, which we standardize to the interval between 0 and 1 by dividing the scores of 1, 2, 3, and 4 by 4, see, e.g., [33], page 77. Examples include “Do you help your children with alleviating their worries?”, “Do you help your spouse cook meals?” These questions are further detailed by the object of help, i.e., the interviewees were asked whether they were helping with cleaning the house their children, or their spouse, or even their parents. By combining the older individuals’ answers to all these questions, we constructed a combined score in this category by computing an arithmetic average of the scores obtained on the individual questions. The combined indicator for the second domain is computed as an unweighted average of the four component indicators. The weight of the Social Participation domain in the HAAI is 19%.

#### 3.4.3. Independent and Healthy Living

In domain 3 (Independent and Healthy Living) we replaced the original “relative median income” indicator with the “relative income” indicator by computing the ratio of a respondent’s household yearly income per member of household to the median income of individuals aged between 18 and 65 in Korea in 2022 as reported in [34].

We excluded the indicator “no poverty risk” in the original WHO framework because it is computed as the share of older individuals whose disposable income falls below one-half of the national median disposable income. At an individual level this indicator would be computed as a ratio of an older respondent’s income to one-half of the national median disposable income, which is exactly twice the “relative median income” indicator.

Our proxy for the original indicator “No severe material deprivation” is based on a categorical question in our survey that asks about the most problematic item of expenses that includes the following items: (1) rent, mortgage or utility bills (2) heating (3) unexpected expenses (4) meat or proteins (5) holidays (6) television set (7) washing machine (8) car and (9) telephone. We set the value of the “No severe material deprivation” indicator equal to unity in case the respondent reports a problem paying for any of the abovementioned items.

The “Physical safety” indicator reflects the extent to which older individuals feel safe in their area. The related question is “How safe do you feel your area is?” with the answers given on a Likert scale. We standardize these answers to the interval between zero and one. The “Lifelong learning” captures attendance by the survey respondents of courses, seminars, private lessons or other learning activities outside of the formal educational system. The related question is “Have you taken part in any educational activities during the past one year?” The related educational categories include health-related instruction, culture-related courses, foreign language study and the like with the total of six categories. The third domain enters the HAAI with a weight of 21%.

#### 3.4.4. Capacity and Environment for Healthy and Active Ageing

Finally, in domain 4 (Capacity and Environment for Healthy and Active Ageing) we left out the original indicator named “Remaining Life Expectancy at Age 65” since we are unable to compute the respondents’ life expectancy based on their answers to the survey questions. Our indicator 4.1 (Ease of daily living activities) is a replacement of the original indicator 4.2 (Share of Healthy Life years in the Remaining Life Expectancy). We follow the suggestions in [11] to calculate it as an average score on the variety of daily living activities reported by the respondents and three reverse scores related to difficulties in hearing, vision, and chewing.

We capture the “mental well-being” indicator by an average score of the group of questions related to depression. This group includes questions, such as “Have you recently experience a decrease in your eagerness to live?”, same for the feelings of emptiness, boredom, anxiety and the like. This group contains fifteen questions in all.

In our survey we have a group of 11 questions titled “Using electronic devices” that includes ICT-related questions such as “Are you using your phone to send and receive messages?” or “Are you using electronic devices to search for information such as the news?” We take arithmetic average of the binary responses to these questions as a proxy for indicator “Use of ICT”.

Several questions deal with the older individuals’ social connectedness such as “How often do you get in contact with friends or relatives?” The answers to these questions are Likert-scale type. We standardized these answers for each question in this group and took their arithmetic average to be a proxy for indicator “Social Connectedness”. Finally, the last indicator in domain 4 is based on a categorical question that asks about the educational attainment of the survey respondents. The answers vary from “no education at all” to “postgraduate degree” with the total of eight educational levels. The fourth domain enters the HAAI with a weight of 32%.

A detailed description of the questions used to compute each indicator listed in Table 2 along with the coding of answers is available from the authors immediately upon request.

### 3.5. Tobit and Logit Regression Analyses

We compute and analyze the composition of the HAA index and run a series of logit and probit regressions in order to establish the association of the values of the HAA index with a set of physical, lifestyle, and socio-economic determinants. Since both the four dimensions of our index of healthy and active ageing, and the HAAI itself are varying within the range between 0% and 100%, we estimate a series of probit models in order to study the association between the dimensions of healthy and active ageing and a set of their determinants.

Denote Y the value of healthy and active ageing index, $Y\in \left[0\%,100\%\right]$, ${\vec{X}}^{\prime }$ a vector of its determinants, and $\vec{\beta }$ a vector of corresponding coefficients. A conventional linear regression model of the type $Y_{i}={\vec{X}}_{i}^{\prime }\vec{\beta }+\varepsilon_{i}$ where $\varepsilon_{i}$ are independently and identically distributed (i.i.d.) random shocks may result in the predicted values of ${\hat{Y}}_{i}$ that fall outside the range of [0%,100%]. One approach to deal with this problem is to consider a latent variable $Y_{i}{}^{*}={\vec{X}}^{\prime }{\vec{\beta }}_{i}+u_{i}$ where $u_{i}$ is a random i.i.d. normal and the latent, i.e., unobserved, variable $Y_{i}^{*}$ and the observed HAAI values $Y_{i}$ are linked in the following fashion:(1)$$Y_{i}=\left\{\begin{array}{l} 0\%,\;Y_{i}^{*}\leq 0\%\\ Y_{i}^{*},\;Y_{i}^{*}\in \left(0\%,100\%\right)\\ 100\%,\;Y_{i}^{*}\geq 100\% \end{array}\right.$$

Model (1) is described in [35] and is known as a two-limit Tobit model, an extension of the original Tobin model in [36] analyzing a regression modeling in case the dependent variable is censored from below. We estimate two-limit Tobit models in (1) when analyzing the effects of a set of determinants on dimensions 2, 3, and 4 of the healthy and active ageing index, see Table 2 above, as these dimensions are measured within the interval between 0% and 100%.

Since dimension 1 of the HAAI, “Employment”, is a binary variable assuming the value of one if a senior respondent has a job and zero otherwise, probit would be the more appropriate model, see [35], specified as follows:(2)$$\mathrm{Pr}\left\{Y_{i}=1\right\}=\Phi \left({{\vec{X}}^{\prime }}_{i}\vec{\beta }\right)$$
where $\Phi \left(•\right)$ is a standard normal cumulative distribution function (cdf). In case the right-hand-side in (2) is given by the logistic, rather than the normal, distribution function, model (2) is referred to as logit.

We use Tobit and probit models in (1) and (2) in order to analyze the association between the components of healthy and active ageing index, as well as the index itself, and a set of its determinants, and report the results in Section 4.4.

## 4. Results

### 4.1. Levels of the Healthy and Active Ageing Index

By construction (see Section 3.4) the values of HAAI and its four components vary between zero and 100. The average level of HAAI is a little lower than 50 at the level of 46.87 with no one in our sample characterized by the highest possible value of 100, suggesting significant room for improvement in the quality of ageing in South Korea. Figure 1 demonstrates that this index has a bimodal distribution with the two modes separated by the HAAI value of approximately 50:

Table 3 suggests that being employed is the key factor driving the difference between the two modes with the employed individuals gaining on average 30 points in terms of the HAAI compared to their unemployed counterparts:

As demonstrated by Table 2, the summary statistics are strikingly similar for the HAAI values computed for the employed individuals and the ones whose values of HAAI are above 50, which is roughly the dividing value between the two modes in Figure 1. Similarly, the HAAI characteristics computed for the unemployed senior respondents are practically the same with those individuals whose HAAI values are below 50.

We do not find any other binary variable in our dataset such as, e.g., gender that would so clearly condition the difference between the two modes of the HAAI distribution in Figure 1, implying the importance of employment of senior citizens both in terms of the quality of their ageing and in terms of the corresponding policy implications such as, e.g., the government programs of employing the older citizens.

Further corroborating the importance of being employed to the value of the healthy and active ageing index is the fact that the average values of three other dimensions of this index are greater for the employed compared to the unemployed group of the respondents. Table 4 presents the results of the *t*-tests comparing the average values of these three dimensions based on the employment status.

In case of all three sub-dimensions the employed respondents are characterized by higher values of the corresponding component of the HAAI with the difference being statistically significant. Those senior individuals who report having a paid job are also characterized by a higher score in terms of their social participation, healthy living, and the extent to which their living environment is conducive to healthy and active ageing compared to their unemployed counterparts.

### 4.2. Composition of the Healthy and Active Ageing Index

In Table 5 below we provide summary statistics on the level and composition of the healthy and active ageing index (hence HAAI).

Table 4 suggests that overall the index of healthy and active ageing is dominated by the domains of the independent and healthy living, and the capacity and environment. Rather regretfully, the social participation domain appears to be contributing only seven percent to the overall HAAI value. 

In Figure 2 we detail the contributions of each domain to the total value of the HAAI for the total sample of the respondents and separately for the male and female subsamples. Formally, denoting $s_{1},s_{2},s_{3}\;\mathrm{and}\;s_{4}$ to be the individual scores for domains 1 through 4, and $w_{i},i=1.4$ their corresponding weights specified in Table 2 such that $\sum_{i=1}^{4}w_{i}=1$, the HAAI index for each respondent has the form $HAAI=w_{1}s_{1}+w_{2}s_{2}+w_{3}s_{3}+w_{4}s_{4}$. The contribution of domain *i* then is given by $\frac{w_{i}}{HAAI}$.

As suggested by Figure 2, the HAAI composition is rather similar for both men and women respondents. The Employment domain appears to be contributing marginally more to the overall HAAI value in case of females compared to males, while the female respondents are characterized by a larger contribution of the Capacity and Environment domain compared to male respondents. However, in all cases the two domains, i.e., “Capacity and Environment” and “Independent and Healthy Living” are contributing approximately three quarters to the overall HAAI value.

### 4.3. Summary Statistics of the Determinants of Healthy and Active Ageing

As mentioned in Section 2.2, we consider three groups of the determinants of healthy and active ageing: physical characteristics, lifestyle variables, and the socio-economic characteristics. Table 6 presents these determinants’ summary statistics.

An average senior respondent in our sample is 74 years old with 90% of the respondents being younger than 84. Three individuals are aged 99, 101, and 102. The body mass index computed as a ratio of one’s weight to the square of one’s height in kilograms and meters, respectively, is commonly used as an indicator of obesity with the range between 18 and 25 considered to be healthy, see, e.g., [37]. The BMI of an average older Korean is well within that range, which is probably a consequence of the traditional preference for seafood and vegetables, with only a little over one percent being obese with the BMI of over 30. Sadly, 90% of the respondents report having been diagnosed with four or more chronic diseases, such as, e.g., diabetes or asthma with the average number of such diseases being approximately six. Females and males are equally well represented by the sample.

Most of the respondents reported controlling their nutrition (84%) and consuming a sufficient amount of fruits and vegetables (90%). Approximately one-half of the senior individuals (48%) are regularly exercising. Thirty-seven percent of the respondents reported no consumption of alcoholic beverages during the last one year. However, more than one-half of the remaining senior individuals reported having at least two or three occasions per month when they drink alcohol. Consumption, however, stays moderate at an equivalent of approximately two pints of beer for those who drink at all. It is worthwhile noting at this point that Korean working culture views regular pastime with colleagues after the working hours as an important way of strengthening ties within the firm, which often involves consuming alcohol. Approximately 10% of the older respondents referred to themselves as smokers. Rather disappointingly, a relatively small share of the older respondents have reported traveling (25%) or attending educational programs or courses (11%) during the past one year.

A little less than a third of the individuals in our sample have graduated from high school with almost one-half of the respondents (45%) having only had an elementary school education. In contrast, only approximately five percent of the older individuals have graduated from a university. This is an important observation in light of the much higher rates of college graduates in modern Korea reaching up to 70% for the age group between 25 and 34. The average income of an older individual in our sample relative to the median income of a person aged 18 to 65 is approximately 40% with only 5% of senior citizens enjoying more than the median income of the younger generation. More than one-half of the older respondents (60%) reported having a spouse.

While a very small share of the respondents (2.6%) report having a working pension income, a much larger share (70.4%) says they are benefiting from a basic pension plan. Similarly, 30.4% report having a public pension income. It is important to keep in mind that these pension incomes are not mutually exclusive. Most respondents (87%) report living either in a townhouse (40%) or in an apartment building (47%) with the rest thirteen percent residing in standalone houses or dwellings of other types. Even if we make a distinction in our analysis between urban and rural areas, it is worthwhile noting that apartment complexes are a regular feature of Korea’s rural areas due to its high extent of urbanization. Thus, even if approximately a third of our respondents are formally living in Seoul, Busan, or other major cities, life in many provincial counties shares many characteristics with life in the officially urban areas.

### 4.4. Statistical Analysis of the Determinants of Healthy and Active Ageing

In Table 7 we present estimation results of the logit and probit models discussed in Section 3.5 for both the individual dimensions of the index of healthy and active ageing, and the aggregate HAAI. The components of individual dimensions of the HAAI are explained in Section 3.4, Table 2. We discuss the implications of these empirical results in Section 5.

All models do significantly better than empty models according to the Chi-squared criterion. In general, the physical, lifestyle, and socio-economic characteristic are estimated to matter for both the aggregate healthy and active ageing index, and its individual dimensions, although the direction and strength of the effects produced by these characteristics is not always uniform across these dimensions. In the next section, we discuss the implications of the results presented in Table 7.

## 5. Discussion

### 5.1. Physical Characteristics

Age, body mass index and the number of chronic diseases are expectedly negatively associated with the dimensions of healthy and active ageing in most cases. Statistical significance of the negative coefficient on BMI squared implies the existence of a “threshold” BMI value beyond which gaining weight becomes detrimental to the value of one’s HAAI. The average of this “threshold value” equals approximately 25, which is the upper limit of the WHO’s definition of non-obesity, see, e.g., [37]. Approximately 19% of the individuals in our sample have the BMI values exceeding 25, indicating possible obesity problems in one-fifth of Korea’s senior population. 

The negative association between age and the quality of ageing is not surprising as age is taxing in terms of one’s ability to find employment or in terms of the constraints it imposes on one’s capacity to continue to age in a healthy and active fashion. We do not have, however, a ready explanation for a positive association between age and the “Healthy and Independent Living” dimension. Our preliminary guess would be that, as the *t*-tests suggest, the group of older individuals living separately from their children is approximately 1.5 years older than the rest, probably making one’s age associated with more independent living. Furthermore, the values of the ‘relative income’, ‘material deprivation’, ‘physical safety’ and ‘lifelong learning’ indicators are likely to be higher for the older people. Thus, one study on the effect of education for the elderly on their life satisfaction suggests that social relationships can be formed through the educational activities, generating a sense of fulfillment and overall satisfaction with life, see [38]. An important consequence is that those elderly individuals who participate in the employment-related programs are also using these programs as a means of forgetting their livelihood problems and loneliness, thus contributing to an improvement along the dimensions of ‘material deprivation’, ‘lifelong learning’ and ‘physical safety’.

However, more research is needed in this area in order to arrive at a definitive conclusion.

The number of chronic diseases is associated with the lower quality of ageing whenever statistically significant. As mentioned in Section 2.1 on the literature review, the philosophy behind the healthy and active ageing index is to treat chronic diseases as inputs, rather than outcomes, of the ageing process. From that perspective, the presence of a number of chronic illnesses per se is not necessarily conducive to one’s inability to do physical exercise, live separately from one’s children, enjoy an acceptable level of income, or perform well on any one of the seven components of the “Independent and Healthy Living” dimension. However, our results suggest that the total effect of chronic diseases on healthy and active ageing is negative.

A rather discouraging finding is that, except for the “Social Participation” dimension, being female is associated with a lower probability of being employed, lower scores on the “Independent and Healthy Living” and “Capacity and Environment” dimensions, and a lower value of the HAAI in general. Thus, according to the Korean Development Institute report, see [7], the labor force participation rate in Korea among senior citizens older than 65, is twice as low (24.1%) compared to that of men (41.5%). In our sample, 30.4% of the female respondents reported to have a paid job, while the share of employed males stood at 47.7%. Encouraging females’ employment, especially at an older age, appears to be an important government policy area. The effectiveness of this policy, however, may be limited due to the fact that Korea’s labor participation rate for both men and women is already higher than that in the G7 group, see Table 2 in [7].

### 5.2. Lifestyle Characteristics

Not all lifestyle characteristics were included into the list of all of the dimensions’ determinants to avoid problems of endogeneity. Thus, traveling is excluded for the “Employment” dimension because being employed is typically associated with a higher income, thus increasing one’s ability to travel. Similarly, physical exercise is not listed as a determinant of the “Independent and Healthy Living” dimension since physical exercise is part of the definition of the dimension itself.

The estimation results in Table 7 suggest that drinking more frequently and smoking among senior respondents appears to be positively associated with the probability of being employed as well as with a higher HAAI score in general. These results can be explained by a relationship between subjective social support and drinking. Subjective social support refers to all the positive resources that an individual can derive from his or her interpersonal relationships. Through the exchange of emotional attention, comfort, practical help, and understanding between people, individuals can experience a feeling of being cared for, loved, valued, and belonging to a network of conversations and relationships. Subjective social support has been reported to have the ability to improve psychological satisfaction and physical health in various ways, see [38].

When it comes to drinking, [39] found that groups of individuals with a high extent of subjective social support tend to drink more alcohol compared to the groups characterized by a low extent of social support. The authors explain the relationship by mentioning that drinking alcohol plays a mediating role in forming the interpersonal relationships, especially when negative life events occur. Additionally, [40] find that the social support networks formed while sharing alcoholic drinks produces a significant effect on the quality of the relationships between individuals.

Several studies such as, e.g., [41] provide evidence that Korean employees tend to release their stress by alcohol drinking. An important source of the work-related stress among seniors is that, as emphasized in [7], senior employees are generally choosing to work in order to make ends meet rather than out of the desire to age actively, thus ending up in the pool of low-quality employment. In addition, relatively widespread is the culture of hoesik, or after-work drinking sessions among the employees with the purpose of sharing ideas, building social networks or fostering team spirit, see [42] for details. Drinking alcohol thus can be considered as a consequence of being employed. Three-quarters of the employed respondents in our sample report drinking alcohol two to three times every week. A positive relationship between smoking and employment is probably of the same nature with the one between drinking and employment.

While a positive association between physical exercise and the “Capacity and Environment” dimension is hardly surprising, we also discover a negative association between physical exercise and employment. As mentioned above, senior employment in Korea is likely to be associated with the job-related stress, which in turn was found to result in a lower level of physical activity, see, e.g., [43]. In this case more physical activity may be indicative of an absence of job-related stress because of the absence of the job itself.

Interestingly, certain lifestyle characteristics such as the quality of nutrition and lifelong learning do not appear to be important for healthy and active ageing as measured by the HAAI. This is especially worrying in case of the lifelong learning because of the positive effect produced by senior citizens’ educational activities on their ability to age in an “empowered and liberated” fashion, as formulated by [44].

### 5.3. Socio-Economic Characteristics

Expectedly, a higher relative income contributes positively to both “Social Participation” and “Capacity and Environment” dimension. A positive association between the level of education and the dimensions of “Social Participation” and “Independent and Healthy Living” is not surprising either. It is the statistically significant and negative association between education and the probability of being employed that requires an explanation.

As mentioned above, senior Korean citizens are most likely seeking employment in order to make ends meet. Since a higher level of one’s life-time educational achievement is indicative of the quality of one’s job held prior to entering the ranks of senior citizens and, as a result, of a lesser need to “make ends meet”, e.g., due to a higher working pension income. The unemployed senior respondents in our sample turn out to be mostly the graduates of elementary or middle school, while the employed pool predominantly consists of the high school graduates with the difference in the education level being statistically significant according to a two-sample t-test. We believe this finding underscores yet again the problem of senior employment in Korea as a means of “making ends meet” rather than an expedient of ageing in an active and healthy way.

We estimate marriage status to be positively associated with the “Social Participation” domain and the HAAI value in general, while observing a negative association with the employment probability. Since marriage is generally conducive to more financial stability, it may act as a disincentive to search for employment given the latter’s status of a means to escape from poverty.

Receiving pension is mostly negatively associated with the dimensions of healthy and active ageing. The coefficient on the basic pension dummy, for instance, is always statistically significant and negative, including the employment dimension. The working pension coefficient is estimated to be positive and statistically significant only in case of the “Social Participation” and “Capacity and Environment” dimensions. The probability of employment is negatively associated with all three types of pension, suggesting that higher income resulting from pension payments discourages senior citizens from seeking employment. We believe this finding is of utmost importance as it emphasizes the role of senior employment as a way of escaping poverty rather than a means of contributing to the society in an active fashion.

Except for the “Social Participation” dimension, the number of children living together with the senior respondents is negatively associated with the active and healthy ageing dimensions, possibly reflecting the fact that the senior respondents’ time and energy are diverted from the four dimensions of healthy and active ageing in case one or more children are sharing the same living space with their parents. Finally, living in two of Korea’s major cities such as Seoul and Busan is negatively associated with the healthy and active ageing dimensions, which might reflect, among other things, a higher extent of labor market competition among the seniors seeking employment. Alternatively, considering that residence is a concept that encompasses physical characteristics such as housing and amenities, social characteristics such as bonding with residents, and psychological characteristics such as belonging and attachment to the community [44], the differences between metropolitan, non-metropolitan, urban and rural areas can be important factors affecting the elderly’s life satisfaction. Relevant studies in this area include [45] and [46], and imply that the extent of life satisfaction of the rural elderly is higher than that of the urban elderly, which helps to explain our empirical results. Another important influence is that, since younger generation tends to concentrate in the cities, the older respondents in rural areas are more likely to report living alone, which increases their score on the “Independent and Healthy Living” dimension.

### 5.4. Study Limitations

We understand it would be desirable to use a panel dataset covering the period between 2008 and 2020 rather than a cross-section of senior citizens interviewed in 2020. However, there are two reasons why we decided to limit our analysis to the data from 2020. First, surveys in different years contain information on different individuals so the same person may not be necessarily interviewed in all years. Further, even if he or she took part in several surveys, the survey materials do not contain any reference to the identity of the respondents, which makes the construction and analysis of a panel dataset an impossible task. Second, the focus of this study is on the construction of the index of healthy and active index per se and the analysis of the effects produced on it by the various determinants rather than on the investigation of the dynamics of these influences.

## 6. Conclusions

In this study, we used the results of a unique National Survey of Older Koreans conducted in 2020 and covering more than ten thousand senior citizens in Korea in order to construct an individual-level measure of healthy and active ageing and to estimate its association with the senior respondents’ physical, lifestyle, and socio-economic characteristics.

The basis for our index of healthy and active ageing (HAAI) is the WHO policy framework for the evaluation of active ageing at an aggregate level as documented in [12]. Using responses from the survey, we modify the WHO methodology in order to adapt the aggregate index of healthy ageing to an individual level. The resulting HAAI values are a weighted average of the four dimensions, such as “Employment”, “Social Participation”, “Independent and Healthy Living”, and “Capacity and Environment” with the weights recommended by a panel of experts and reported in [11].

In total, 90% of older respondents’ HAAI scores are between 30 and 70 with a median value of 40, suggesting significant room for improvement. The distribution of HAAI values is bi-modal, with the higher mode corresponding to employed individuals. The dimensional composition of the HAAI is similar between men and women with the “Social Participation” and “Independent and Healthy Living” dimensions contributing the most to the overall value of the index of healthy and active ageing.

In order to analyze the association between the various dimensions of HAAI and its physical, lifestyle, and socio-economic determinants, we run logit and probit regressions with the HAAI’s four dimension scores as dependent variables. We believe our most interesting results are those related to the probability of being employed, which is one dimension of the healthy and active ageing index. Thus, we were surprised to find a positive association between alcohol consumption and employment, as well as a negative association between all types of pension and the probability of being employed. As we argue in Section 5, an important role in interpreting our empirical results is played by the apparent role of Korea’s senior employment as a means of “making ends meet” rather than a way of improving the quality of one’s ageing. This phenomenon is discussed, among others, by the Korea Development Institute policy report in [7]. We believe it is because of this characteristic of Korea’s senior employment that more education and less physical activity, for instance, is associated with a lower probability of being employed. Similarly, we argue that a positive association between alcohol consumption and employment is likely to be explained by work-related stress, which would characterize employment as a necessity as opposed to employment as a means of self-realization. A negative association between receiving pension income and employment is also explained within this paradigm as more pension income makes stressful, or menial, employment less desirable.

Given the relatively low contribution of the “Employment” dimension to the overall value of the healthy and active ageing index, and the likely characteristic of Korea’s senior employment as a means to “make the ends meet”, we believe an important policy implication would be to promote higher-quality employment opportunities for senior citizens. This type of policies is especially important as Korea’s labor force participation rate among senior citizens is already high compared to G7 countries, such as, e.g., Japan or Sweden, see [7], so encouraging senior employment per se is not likely to produce a positive effect on the quality of ageing.

## Figures and Tables

**Figure 1 ijerph-19-16802-f001:**
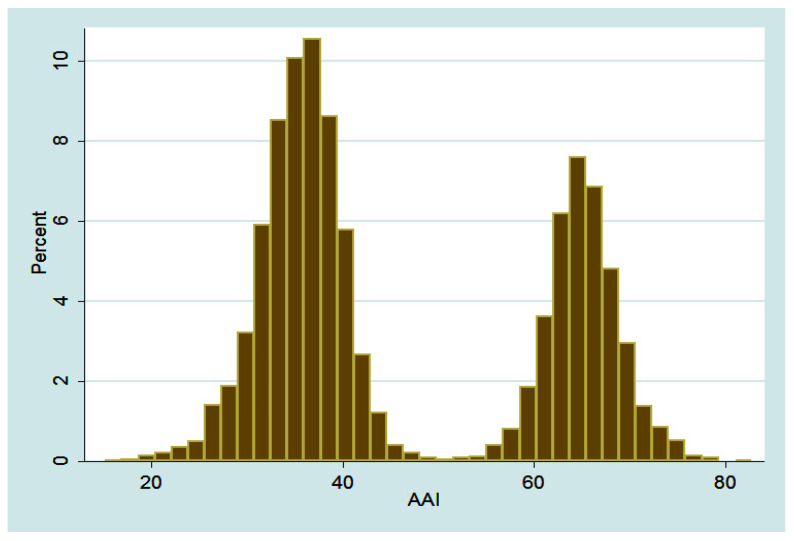
Distribution of the Values of the Healthy and Active Ageing Index.

**Figure 2 ijerph-19-16802-f002:**
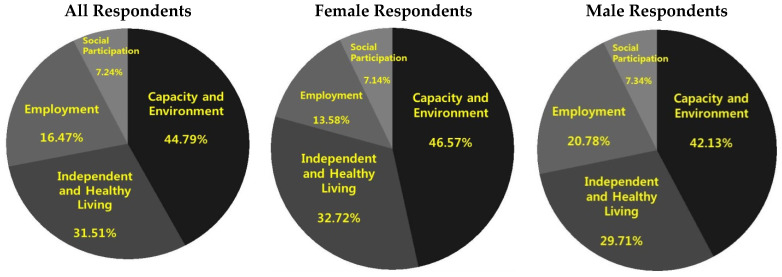
Components of the Healthy and Active Ageing Index.

**Table 1 ijerph-19-16802-t001:** Population and Sampling Characteristics of the Korean Elderly Survey.

Administrative District	Population Status of the Elderly Aged 65 and over (Population)			Sampling Result		
	Dong Area	Eup/Myeon Area	Total	Dong Area	Eup/Myeon Area	Total
**Korean Total**	5,439,457	1,954,658	7,394,115	7140	2860	10,000
**Seoul**	1,335,559		1,335,559	1070	0	1070
**Busan**	572,642		572,642	720	0	720
**Daegu**	356,842		356,842	570	0	570
**Incheon**	352,691		352,691	570	0	570
**Gwangju**	188,115		188,115	420	0	420
**Daejeon**	186,661		186,661	420	0	420
**Ulsan**	120,578		120,578	400	0	400
**Sejong**	27,705		27,705	200	0	200
**Gyeonggi**	1,166,739	340,354	1,507,093	870	260	1130
**Gangwon**	139,453	139,839	279,292	250	260	510
**Chungbuk**	114,702	139,639	254,341	220	270	490
**Chungnam**	97,392	263,362	360,754	150	420	570
**Jeonbuk**	184,685	162,907	347,592	300	260	560
**Jeonnam**	105,344	285,487	390,831	160	440	600
**Kyeongbuk**	186,491	326,795	513,286	250	440	690
**Kyeongnam**	247,033	260,232	507,265	330	350	680
**Jeju**	56,825	36,043	92,868	240	160	400

*Note: Dong (동; 洞) is the primary division of districts (gu), and of those cities (si) that are not divided into districts. The dong is the smallest level of urban government. Eup (읍; 邑) is similar to the unit of town. Along with myeon, an eup is one of the divisions of a county (gun), and of some cities (si) with a population of less than 500,000 people. myeon (면; 面) is one of the divisions—along with eup—of a county (gun) and some cities (si) of fewer than 500,000 population. Myeons have smaller populations than eup and represent the rural areas of a county or city. Source: [15]*.

**Table 2 ijerph-19-16802-t002:** The Individual Healthy and Active Ageing Index.

Domains			
Actual Ageing Experience			Potential to Age Actively
1. Employment 28%	2. Social Participation 19%	3. Independent and Healthy Living 21%	4. Capacity and Environment for Healthy and Active Ageing 32%
1.1 Employment 100%	2.1 Voluntary activities 19%	3.1 Physical exercise 2.3%	4.1 Ease of daily living activities 34.9%
	2.2 Care to children and grandchildren 46%	3.2 Access to health and dental care 29.9%	4.2 Mental well-being 30.2%
	2.3 Care to older adults 22%	3.3 Independent living 27.6%	4.3 Use of ICT 6.3%
	2.4 Political participation 13%	3.4 Relative income 13.8%	4.4 Social connectedness 19.1%
		3.5 No severe material deprivation 14.9%	4.5 Educational attainment 9.5%
		3.6 Physical safety 10.3%	
		3.7 Lifelong learning 1.2%	

*Source: adapted for computation at an individual level from [11]. Percentage values represent weights of the domains and their sub-domains.*

**Table 3 ijerph-19-16802-t003:** The Levels of Healthy and Active Ageing Index.

	Mean	Standard Deviation	Minimum	Maximum
**Aggregate HAAI**	46.87	15.05	15.20	82.81
**HAAI for Employed**	65.25	3.87	49.38	82.81
**HAAI for AAI > 50**	65.25	3.87	50.13	82.81
**HAAI for Unemployed**	35.49	4.32	15.20	50.13
**HAAI for AAI < 50**	35.49	4.32	15.20	49.37

**Table 4 ijerph-19-16802-t004:** T-tests Comparing the Values of the HAAI Components depending on the Employment Status.

	Mean	Standard Deviation	Minimum	Maximum
**Employment Status**	**Social Participation**			
**Employed**	18.01	9.49	0	72.21
**Unemployed**	15.67	8.19	0	61.71
**Mean Difference**	−2.34 (−13.09) ***	8.77		
	**Healthy and Independent Living**			
**Employed**	65.18	12.40	2.41	93.83
**Unemployed**	63.81	13.16	5.22	93.83
**Mean Difference**	−1.37 (−5.14) ***			
	**Capacity and Environment for Healthy and Active Ageing**			
**Employed**	62.47	6.73	24.70	94.89
**Unemployed**	59.24	7..94	16.87	87.45
**Mean Difference**	−3.23 (−20.20) ***			

*Note: Mean difference for each sub-dimension of the HAAI equals the difference between the mean value for the employed minus the mean value for the unemployed sub-sample. T-statistics are in parentheses. *** refers to statistical significance at a 1% level.*

**Table 5 ijerph-19-16802-t005:** Individual Components of the Healthy and Active Ageing Index in Korea.

	Mean	Standard Deviation	Minimum	Maximum
	**Levels**			
**Employment**	37.35	48.38	0	100
**Social Participation**	16.55	8.77	0	72.21
**Independent and Healthy Living**	64.33	12.90	2.41	93.83
**Capacity and Environment**	60.47	7.66	16.87	94.89
	**Shares**			
**Employment**	16.47	20.99	0	56.70
**Social Participation**	7.23	3.86	0	28.98
**Independent and Healthy Living**	31.51	9.78	1.03	64.25
**Capacity and Environment**	44.79	12.03	15.28	83.43

**Table 6 ijerph-19-16802-t006:** Summary Statistics of the Determinants of Healthy and Active Ageing.

	Physical Characteristics			
**Variables**	**Mean**	**Standard Deviation**	**Min**	**Max**
**BMI * (body mass index)**	23.56	2.61	13.06	46.08
**Age**	73.58	6.63	65	102
**Chronic Diseases**	5.83	1.47	2	21
	**# females**	**% females**	**# males**	**% males**
**Gender**	6062	59.82%	4072	40.18%
	**Lifestyle Characteristics**			
**Variables**	**Mean**	**Standard Deviation**	**Min**	**Max**
**Drinking frequency**	1.14 (once a month)	1.75	0 (never)	7 (every day)
**Drinking volume**	1.41 (pints of beer)	2.31	0.5 (pint of beer)	11.5 (pints of beer)
	**# No**	**% No**	**# Yes**	**% Yes**
**Smoker**	8993	89.1%	1104	10.9%
**Controls nutrition**	1656	16.4	8441	83.6%
**Consumes sufficient fruits and vegetables**	1034	10.2%	9100	89.8%
**Physical exercise**	5242	51.9%	4855	48.1%
**Traveling** **(last year)**	7555	74.8%	2542	25.2%
**Educational courses (last year)**	8979	88.9%	1118	11.1%
	**Socio-Economic Characteristics**			
**Variables**	**Mean**	**Standard Deviation**	**Min**	**Max**
**Relative Income ****	0.40	0.28	0.01	2
**Educational Achievement**	3.80 (middle school)	1.24	0 (no school)	7 (college and higher)
	**# No**	**% No**	**# Yes**	**% Yes**
**Married**	4203	41.5%	5931	58.5%
**Working pension**	9833	97.4%	264	2.6%
**Public pension**	7029	69.6%	3068	30.4%
**Basic pension**	2985	29.6%	7112	70.4%
**Townhouse**	6125	60.4%	4009	39.6%
**Apartment**	5354	52.8%	4780	47.2%
**Seoul or Busan**	9418	92.9%	716	7.1%
**Urban area**	7514	74.15%	2620	25.85%

*Notes: * BMI is defined as one’s body weight in kilograms divided by the square of one’s height in meters. ** Relative income is the ratio of an older individual’s income to the median income for the age group of 18–65. We report summary statistics for the 99% of the sample whose relative income is less than 2. Some of the above variables are also components of one of the dimensions of the healthy and active ageing index, and thus will not be used in all regression specifications.*

**Table 7 ijerph-19-16802-t007:** Physical, Lifestyle, and Socio-Economic Determinants of Healthy and Active Ageing.

Variables	Employment	Social Participation	Healthy and Independent Living	Capacity and Environment	Healthy and Active Ageing Index
**Model**	**Logit**	**Probit**	**Probit**	**Probit**	**Probit**
	**Physical Characteristics**				
**BMI (body mass index)**	0.203 (2.41) **	0.907 (4.08) ***	−0.232 (−0.81)	1.641 (6.75) ***	1.707 (3.65) ***
**BMI^2^**	−0.004 (−2.21) **	−0.018 (−3.96) ***	0.004 (0.70)	−0.034 (−6.75) ***	−0.034 (−3.53) ***
**Age**	−0.098 (−20.39) ***	−0.048 (−3.80) ***	0.037 (2.29) **	−0.333 (−26.6) ***	−0.580 (−24.56) ***
**Chronic Diseases**	−0.152 (−8.41) ***	−0.046 (−0.95)	0.044 (0.71)	−0.703 (−13.96) ***	−0.944 (−9.74) ***
**Gender**	−0.645 (−10.69) ***	1.063 (6.04) ***	−0.564 (−2.54) **	−0.820 (−4.63) ***	−3.460 (−10.17) ***
	**Lifestyle Characteristics**				
**Drinking frequency**	0.112 (5.27) ***	−0.070 (−1.09)	−0.419 (−5.22) ***	0.197 (2.98) ***	0.601 (4.75) ***
**Drinking volume**	0.016 (0.99)	0.085 (1.75) *	0.003 (0.04)	−0.005 (−0.10)	0.092 (0.96)
**Smoker**	0.360 (4.47) ***	−0.641 (−2.69) ***	−0.312 (−1.04)	0.198 (0.80)	2.434 (5.09) ***
**Controls nutrition**	−0.067 (−1.01)	−0.666 (−3.53) ***	0.203 (0.85)	−0.113 (−0.58)	−0.631 (−1.67) *
**Consumes sufficient fruits and vegetables**	−0.001 (−0.02)	−0.661 (−2.85) ***	1.831 (6.23) ***	0.296 (1.21)	−0.325 (−0.69)
**Physical Exercise**	−0.310 (−6.39) ***	0.062 (0.45)		1.531 (10.67) ***	
**Traveling** **(last year)**		0.983 (5.96) ***	0.565 (2.74) ***	1.861 (11.08) ***	
**Educational courses (last year)**	−0.065 (−0.85)	−0.234 (−1.09)		0.058 (0.26)	
	**Socio-Economic Characteristics**				
**Relative Income**		0.287 (2.40) ***		0.317 (2.66) ***	
**Educational Achievement**	−0.115 (−4.70) ***	0.562 (8.07) ***	0.944 (10.82) ***		
**Married**	−0.222 (−4.09) ***	10.157 (64.92) ***	0.231 (1.18)	0.051 (0.31)	0.973 (3.16) ***
**Working pension**	−1.559 (−9.33) ***	1.008 (2.34) **	0.645 (1.20)	1.937 (4.39) ***	−7.417 (−8.76) ***
**Public pension**	−0.865 (−10.04) ***	−0.470 (−1.83) *	0.260 (0.80)	0.037 (0.14)	−5.232 (−10.30) ***
**Basic pension**	−1.375 (−21.33) ***	−1.079 (−5.64) ***	−0.956 (−4.03) ***	−1.014 (−5.13) ***	−8.792 (−23.57) ***
**Number of children living together**	−0.206 (−3.44) ***	5.723 (34.73) ***	−23.417 (−111.71) ***	−0.529 (−2.81) ***	−4.921 (−13.74) ***
**Townhouse**	0.040 (0.55)	0.203 (0.93)	0.510 (1.86) *	0.768 (3.40) ***	0.657 (1.51)
**Apartment**	−0.594 (−8.15) ***	0.137 (0.64)	0.450 (1.68) *	0.957 (4.33) ***	−3.221 (−7.58) ***
**Seoul or Busan**	−0.715 (−7.15) ***	−0.676 (−2.53) **	−2.231 (−6.68) ***	0.217 (0.78)	−4.207 (−7.89) ***
**Urban area**	−0.380 (−6.78) ***	−0.866 (−5.49) ***	0.091 (0.46)	0.708 (4.32) ***	−2.065 (−6.57) ***
**Constant**	7.337 (6.52) ***	1.076 (0.36)	64.054 (16.57) ***	67.671 (21.08) ***	84.102 (13.64) ***
**No. Obs.**	10,079	10,079	9905	9306	9306
**Log-Likelihood**	−5438	−33447	−35149	−30876	−36972
**Log-Likelihood Ratio Chi^2^**	2456 ***	5400 ***	8466 ***	2557 ***	2926 ***

*Note: z-values are in parentheses. Statistical significance levels: *** 1%, ** 5%, * 10%.*

## Data Availability

The dataset used in this study was obtained from the National Survey of Older Koreans conducted by the Korean National Institute of Health and Social Affairs.

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
