# Peer review of "Determinants of Healthy and Active Ageing in Korea"

_ijerph, 2022, doi:10.3390/ijerph192416802_

Round 1

Reviewer 1 Report

manuscript comments

I have read the manuscript determinants of healthy and active aging in Korea, I think it is an interesting work, very well written and with an appropriate bibliographical discussion, I only have some suggestions that I detail below. I suggest not to put abbreviations that are not used during the text, for example, TAPI.

Abstract: do not use the WHO abbreviation that is defined later in the text.

Pag1 1. Line 24: correct fertility

Page 3. Line 11, WHO is said above (page2, line 62)

Page 4 line 158, HAAI is repeated

Page 5, line 225, answer

Page 5, formula 1: it is not clear to me why p=0.37 is used since it is intended to measure other proportions and not the percentage of senior citizens in korea.

Page 8. Line 341: I am not sure if the standardization carried out is correct, explain more about it and give references to where it has been done previously, since it is a discrete variable that is placed on the scale [0,1].

Page 12. Line 479: improve the paragraphs of this line and the next, the statement made is not clear to me.

Page 13: figure 2: it is not clear to me how it was done, explain better.

Author Response

A reply to Referee 1’s comments on

“ Determinants of Healthy and Active Ageing in Korea

 First off, we would like to thank Referee 1 for his or her very useful comments that we believe have helped to substantially improve our paper. Please find below our detailed response. The comments of Referee 1 are cited in bold italicsour response is in plain font. We cite ourselves in italics. Page and paragraph numbers refer to the revised version.

I have read the manuscript determinants of healthy and active aging in Korea, I think it is an interesting work, very well written and with an appropriate bibliographical discussion, I only have some suggestions that I detail below. 

We appreciate your positive perception of our work. Please find our detailed response to your comments below.

I suggest not to put abbreviations that are not used during the text, for example, TAPI.

We removed the TAPI abbreviation. We also got rid of KIHSA and IRB in the first two paragraphs of Section 3.1. We also removed the NSO acronym.

Abstract: do not use the WHO abbreviation that is defined later in the text.

 We modified the first sentence of the abstract in the following way: Based on a framework developed by the World Health Organization, we construct an individual-level percentage measure of healthy and active ageing employing the results of a unique survey of ten thousand elderly Korean respondents conducted in 2020 and relate its values to the senior respondents' physical, lifestyle, and socio-economic characteristics.”

Pag1 1. Line 24: correct fertility

It was not us who wrapped fertility from one line to another so it became “fer-tility, it is the MDPI template that did it. We are afraid we cannot do much about it

Page 3. Line 11, WHO is said above (page2, line 62)

We removed the WHO acronym from the abstract so that the WHO acronym appears and is deciphered on line 64 for the first time in the manuscript.

Page 4 line 158, HAAI is repeated

We removed the HAAI acronym.

Page 5, line 225, answer

Same as “fertility” wrap on line 24it is the template, not us

Page 5, formula 1: it is not clear to me why p=0.37 is used since it is intended to measure other proportions and not the percentage of senior citizens in korea.

Indeed, p is a ratio of the number of older citizens in Korea in the total number of households, as specified in Lee et al. (2020), page 53. However, at the suggestion of one of the referees we removed the technical part containing this discussion altogether.

Page 8. Line 341: I am not sure if the standardization carried out is correct, explain more about it and give references to where it has been done previously, since it is a discrete variable that is placed on the scale [0,1].

In lines 337 to 339 we modify the sentence mentioning standardization in the following way: “The answers are Likert-type on a scale from 0 to 4, which we standardize to the interval between 0 and 1 by dividing the scores of 1, 2, 3, and 4 by four, see e.g. [34], page 77.” 

We rescaled this score in a way similar to e.g. Hair et al. (2010), page 77. We add a corresponding reference on line 379.

Page 12. Line 479: improve the paragraphs of this line and the next, the statement made is not clear to me.

The main idea is that the employed senior individuals score higher on the HAAI dimensions compared to their unemployed counterparts. We add a clarifying sentence on lines 477-480: “Those senior individuals who report having a paid job are also characterized by a higher score in terms of their social participation, healthy living, and the extent to which their living environment is conducive to healthy and active ageing compared to their unemployed counterparts.”

Page 13: figure 2: it is not clear to me how it was done, explain better.

As we explain in Section 3.4, the HAAI index is a weighted average of the four scores related to the four dimensions, namely, employment (1), social participation (2), independent and healthy living (3), and potential to age actively (4). Each domain carries a percentage weight with the sum of the weights equal to 100%. More formally, denoting  their corresponding weights such that . The contribution of domain i then is given by Hair, Joseph. F., William C. Black, Barry J. Babin, and Rolph E. Anderson. 2010. Multivariate Data Analysis. 7th ed. New Jersey: Pearson.

Lee, Yoon Gyeong et al. 2020. 2020 National Survey of Older Koreans. Seoul, Korea: Korean institute for Health and Social Affairs. Policy Report.

Reviewer 2 Report

Thank you for inviting me as a reviewer of this valuable manuscript. I recommend following suggestions for improving the quality of manuscript.

(Comment 1) Overall, this paper is too detailed in certain parts (literature review, methods and results). Because the presentation of this paper is not well-organized, it is difficult to understand the purpose, main results, interpretation of results, and policy contributions. I recommend authors to re-write the manuscript referring previous studies.

Introduction

(Comment 2) I recommend authors to add references for following sentences.

- (line 26-30) It took Korea only seventeen years to become an aged society with over 14% of her population older than 65 in 2017 as opposed to Japan where the same transition took twenty-four years [ref]. Meanwhile Korea's TFR fell to 0.81 in 2021, the world's lowest level [ref]. In the year of 2020 Korea has officially registered a first-ever decrease in its population [ref].

- (line 31-33) Four regions in Korea have already reached the super-aged status, i.e. the provinces of North and South Jeolla, North Gyeongsang, and Gangwon [ref].

- (line 36-38) Increased coverage of Korea's older population by medical insurance and improved general living standards resulted in a significant increase in the older generation's life expectancy [ref].

- (line 41-42) Korea's ageing population brings forth a host of well-known problems such as e.g. a shrinking labor force or a higher dependency ratio [ref].

- (line 44-49) Reduced tax revenues due to the shrinking labor force will not only result in lower spending on the welfare programs for the elderly, but also in the pension fund deficits, dealing another blow to the well-being of the older generation [ref]. The effects of ageing are likely to contribute to a decrease in Korea's economic growth rate that is projected to fall to the level of 1% by 2049 by the Korea's Development Institute, potentially creating a vicious circle in the Korean economy [ref].

- (line 50-57) Korea's population ageing necessitates the government’s policy response such as e.g. developing fiscal policies aimed at establishing healthcare models that are specially designed for the older individuals or policies promoting productivity improvements that are necessary to make up for the rapidly declining labor force [ref]. Also necessary is the creation of opportunities for the older citizens’ productive contribution to the society such as e.g. vocational training and lifelong education programs. Immigration-related policies could play a positive role similarly to the one played by the Turkish immigration to Germany or that of the Poles to Belgium in the middle of the twentieth century [ref].

(Comment 3) line 71-87 is proper Method Section rather than Introduction Section. I recommend authors to remove this sentences and re-write it.

(Comment 4) What is the purpose of this study? Also, what contribution is expected of this study? I recommend authors to clarify this point in the Introduction Section.

Method Section

(Comment 5) Authors mentioned National survey said it started in 2008 (line 214). Why didn't the author integrate and analyze all data from 2008 to 2020? Why did you only analyze 2020 data?

(Comment 6) IRB statement is needed in Method Section and Back matter. It is considered that exemption deliberation is necessary through IRB approval.

(Comment 7) I recommend authors to write 'Sampme Selection Process' by integrating and summarizing line 237-280. Also, consider adding a sample selection figure (e.g. doi:10.1136/ bmjopen-2019-033159).

Author Response

A reply to Referee 2’s comments on

Determinants of Healthy and Active Ageing in Korea

First off, we would like to thank Referee 2 for his or her very useful comments that we believe have helped to substantially improve our paper. Please find below our detailed response. The comments of Referee 1 are cited in bold italicsour response is in plain font. We cite ourselves in italics. Page and paragraph numbers refer to the revised version.

Thank you for inviting me as a reviewer of this valuable manuscript. I recommend following suggestions for improving the quality of manuscript.

Thank you for your positive feedback. Please find our detailed response to your comments below.

(Comment 1) Overall, this paper is too detailed in certain parts (literature review, methods and results). Because the presentation of this paper is not well-organized, it is difficult to understand the purpose, main results, interpretation of results, and policy contributions. I recommend authors to re-write the manuscript referring previous studies.

 In order to increase the readability of the paper, we reduced the amount of text in Section 3 (Materials and Methods). First, in lines 227-243 in the previous version (i.e. the second and the third paragraphs of Section 3.1) the part containing the investigation method and the contents related to post-verification was deleted. Second, a detailed description of the sample size and sample distribution (lines 261-283 in the previous version, i.e. the last two paragraphs of Section 3.2 with formulas (1) and (2)) were deleted as well. Instead, we presented a description of the distribution status by detailed layer according to the proportional distribution method.

Introduction

(Comment 2) I recommend authors to add references for following sentences.

- (line 26-30) It took Korea only seventeen years to become an aged society with over 14% of her population older than 65 in 2017 as opposed to Japan where the same transition took twenty-four years [(1)ref]. Meanwhile Korea's TFR fell to 0.81 in 2021, the world's lowest level [(2)ref]. In the year of 2020 Korea has officially registered a first-ever decrease in its population [(3)ref].

In lines 28-30 we add the following citations:

Lee (2018), Statics Korea (2022a) and Statics Korea (2022b), see References below.

- (line 31-33) Four regions in Korea have already reached the super-aged status, i.e. the provinces of North and South Jeolla, North Gyeongsang, and Gangwon [(3)ref].

We add the following reference on line 34:

Statics Korea (2022b), see References below.

- (line 36-38) Increased coverage of Korea's older population by medical insurance and improved general living standards resulted in a significant increase in the older generation's life expectancy [[2]ref].

We add the following reference to line 39:

Lee (2021), see References below.

- (line 41-42) Korea's ageing population brings forth a host of well-known problems such as e.g. a shrinking labor force or a higher dependency ratio [ref].

We add the following reference to line 43:

Lee (2021), see References below.

- (line 44-49) Reduced tax revenues due to the shrinking labor force will not only result in lower spending on the welfare programs for the elderly, but also in the pension fund deficits, dealing another blow to the well-being of the older generation [[2]ref]. The effects of ageing are likely to contribute to a decrease in Korea's economic growth rate that is projected to fall to the level of 1% by 2049 by the Korea's Development Institute, potentially creating a vicious circle in the Korean economy [Kim, Jung and Hur 2022ref].

We added the following references to line 48: 

Lee (2021) and Kim, Jung, and Hur (2022), see References below.

- (line 50-57) Korea's population ageing necessitates the government’s policy response such as e.g. developing fiscal policies aimed at establishing healthcare models that are specially designed for the older individuals or policies promoting productivity improvements that are necessary to make up for the rapidly declining labor force [Ahn et al. 2017ref]. Also necessary is the creation of opportunities for the older citizens’ productive contribution to the society such as e.g. vocational training and lifelong education programs. Immigration-related policies could play a positive role similarly to the one played by the Turkish immigration to Germany or that of the Poles to Belgium in the middle of the twentieth century [Lee et al. 2015ref].

We add the following references to lines 54 and 59: Ahn, Kim, and Yook (2017) and K. Y. Lee et al. (2015).

(Comment 3) line 71-87 is proper Method Section rather than Introduction Section. I recommend authors to remove this sentences and re-write it.

 We remove the technical part from the Introduction and place it at the beginning of Section 3.5 that discusses regression specifications, lines 404-410.

(Comment 4) What is the purpose of this study? Also, what contribution is expected of this study? I recommend authors to clarify this point in the Introduction Section.

Thank you for pointing this out, we indeed omitted an important part that clarifies the importance of our study. We add the following paragraph discussing the purpose and contributions of this study in the Introduction section, lines 74-82, as follows: “To our knowledge, a comprehensive index of healthy and active ageing developed in accordance with the WHO framework has not been computed in Korea so far on the basis of a country-wide national survey sample. One purpose of this study is thus to provide policy makers with a quantitative idea of the structure of ageing in the Korean society that can also be analyzed in an international perspective. Second, we conduct a rigorous quantitative analysis of the physical, lifestyle, and quantitative characteristics of senior individuals that affect the quality of their ageing. We hope the results of this study are valuable not only in terms of the academic contribution but also in terms of their potential applicability to policy making.”

Method Section

(Comment 5) Authors mentioned National survey said it started in 2008 (line 214). Why didn't the author integrate and analyze all data from 2008 to 2020? Why did you only analyze 2020 data?

It would be indeed a great idea to use a panel dataset covering the period between 2008 and 2020 rather than a cross-section of senior citizens interviewed in 2020. However, there are two reasons why we decided to limit our analysis to the data from 2020. First, surveys in different years contain information on different individuals so that the same person may not be necessarily interviewed in all years. Further, even if he or she took part in several surveys, the survey materials do not contain any reference to the identity of the respondents which makes the construction and analysis of a panel dataset an impossible task. Second, the focus of this study is on the construction of the index of healthy and active index per se and the analysis of the effects produced on it by the various determinants rather than on the investigation of the dynamics of these influences.

To reiterate, we consider such a dynamic analysis a very valuable exercise, but we also believe this sort of analysis is beyond the scope of our study. We will, however, definitely pursue this line of research in the future.

(Comment 6) IRB statement is needed in Method Section and Back matter. It is considered that exemption deliberation is necessary through IRB approval.

We added the following paragraph in lines 227-234:

“Since February 2013, for the purpose of protecting research subjects in accordance with the ‘Bioethics and Safety Act’, the establishment of an Institutional Review Board (IRB), an autonomous review body within research institutions, has been mandated in order to review the scientific and ethical validity of research protocols. Accordingly, the 2020 Elderly Survey applied for IRB for research plans and other research-related documents to the Bioethics Committee established within this institution, and after review, IRB approval (Korea Institute for Health and Social Affairs Bioethics Committee (IRB) Review Result Notification No. 2020 -36) was received.”

(Comment 7) I recommend authors to write 'Sampme Selection Process' by integrating and summarizing line 237-280. Also, consider adding a sample selection figure (e.g. doi:10.1136/ bmjopen-2019-033159).

We add the following clarification on lines 244-247 including the information on sample selection in Table 1:

“For the sample distribution by dong/eup/myeon in each city/province, the proportional distribution method based on the number of elderly people aged 65 or older in the population and housing census was applied. The results of sampling distribution by dong/eup/myeon in each city/province are shown in Table 1 below:”

administrative district

Population status of the elderly aged 65 and over (population)

Sampling result

Dong area

Eup/myeon area

Total

Dong area

Eup/myeon area

Total

National

5,439,457

1,954,658

7,394,115

7,140

2,860

10,000

Seoul

1,335,559

1,335,559

1,070

0

1,070

Busan

572,642

572,642

720

0

720

Daegu

356,842

356,842

570

0

570

Incheon

352,691

352,691

570

0

570

Gwangju

188,115

188,115

420

0

420

Daejeon

186,661

186,661

420

0

420

Woolsan

120,578

120,578

400

0

400

Sejong

27,705

27,705

200

0

200

Kyunggi

1,166,739

340,354

1,507,093

870

260

1,130

Gwangwon

139,453

139,839

279,292

250

260

510

Choongbuk

114,702

139,639

254,341

220

270

490

Choongnam

97,392

263,362

360,754

150

420

570

Jeonbuk

184,685

162,907

347,592

300

260

560

Jeonnam

105,344

285,487

390,831

160

440

600

Kyungbuk

186,491

326,795

513,286

250

440

690

Kyungnam

247,033

260,232

507,265

330

350

680

Jeju

56,825

36,043

92,868

240

160

400

* A dong (동; 洞) is the primary division of districts (gu), and of those cities (si) which are not divided into districts. The dong is the smallest level of urban government.An eup (읍; 邑) is similar to the unit of town. Along with myeon, an eup is one of the divisions of a county (gun), and of some cities (si) with a population of less than 500,000. A myeon (면; 面) is one of the divisions – along with eup – of a county (gun) and some cities (si) of fewer than 500,000 population. Myeons have smaller populations than eup and represent the rural areas of a county or city.

* source: Y. G. Lee et al., “2020 National Survey of Older Koreans,” Korean institute for Health and Social Affairs, Seoul, Korea, Policy Report 2020–35, 2020.

We would like to thank Referee 2 once again for his or her very helpful comments on our study.

References

Ahn, B.K., K.H. Kim, and S.H. Yook. 2017. “The Effect of Population Aging on Growth.” Economic Analysis, Bank of Korea 23(4): 1–33.

Kim, J.Y., K.C. Jung, and J.W. Hur. 2022. “Long-Term Economic Growth Rate Prospects and Implications.” KDI Economic Outlook. https://www.kdi.re.kr/research/analysisView?art_no=3421.

Lee, Hyun-Chool. 2018. “Beyond Silver Democracy: A Quest for New Ideas Regarding Representative Democracy with an Emphasis on the Representation of Future Generations.” Journal of Korean Politics 27(2): 85–114.

———. 2021. “Population Aging and Korean Society.” Korea Journal 61(2): 5–20.

Lee, K.Y. et al. 2015. “International Comparison of Immigration Policies.”

Statics Korea. 2022a. “‘ Birth Statistics in 2021’ Press Release (202208-24).” http://kostat.go.kr/portal/eng/pressReleases/1/index.board?bmode=read&bSeq=&aSeq=420358&pageNo=5&rowNum=10&navCount=10&currPg=&searchInfo=&sTarget=title&sTxt= (November 28, 2022).

———. 2022b. “‘2021 Population and Housing Census (Register-Based Census)’  Press Release (202207-28).” http://kostat.go.kr/portal/eng/pressReleases/1/index.board?bmode=read&bSeq=&aSeq=420172&pageNo=6&rowNum=10&navCount=10&currPg=&searchInfo=&sTarget=title&sTxt= (November 28, 2022).

Round 2

Reviewer 2 Report

The authors have suscessfully addressed all my comments and suggestions. I recommend some minor revision as follows;

(Comment 1) Abbreviation 'KDI' was not described in previous sentences. (line 48)

(Comment 2) Regarding Comment 5, I recommend authors to present following sentences (author's reply) as a study limitation in Discussion Section;

- It would be indeed a great idea to use a panel dataset covering the period between 2008 and 2020 rather than a cross-section of senior citizens interviewed in 2020. However, there are two reasons why we decided to limit our analysis to the data from 2020. First, surveys in different years contain information on different individuals so that the same person may not be necessarily interviewed in all years. Further, even if he or she took part in several surveys, the survey materials do not contain any reference to the identity of the respondents which makes the construction and analysis of a panel dataset an impossible task. Second, the focus of this study is on the construction of the index of healthy and active index per se and the analysis of the effects produced on it by the various determinants rather than on the investigation of the dynamics of these influences.

Author Response

We would like to thank Referee 2 once again for his or her very helpful comments on our study. Please find below our detailed response.

Comments and Suggestions for Authors

The authors have successfully addressed all my comments and suggestions. I recommend some minor revision as follows;

(Comment 1) Abbreviation 'KDI' was not described in previous sentences. (line 48)

In line 48, we replaced KDI abbreviation with the Korea Development Institute.

(Comment 2) Regarding Comment 5, I recommend authors to present following sentences (author's reply) as a study limitation in Discussion Section;

- It would be indeed a great idea to use a panel dataset covering the period between 2008 and 2020 rather than a cross-section of senior citizens interviewed in 2020. However, there are two reasons why we decided to limit our analysis to the data from 2020. First, surveys in different years contain information on different individuals so that the same person may not be necessarily interviewed in all years. Further, even if he or she took part in several surveys, the survey materials do not contain any reference to the identity of the respondents which makes the construction and analysis of a panel dataset an impossible task. Second, the focus of this study is on the construction of the index of healthy and active index per se and the analysis of the effects produced on it by the various determinants rather than on the investigation of the dynamics of these influences.

We also added section 5.4 on the limitations of our study as suggested by the second comment of referee 2.
